# OpenReview forum: "GATv2-NS3 Hybrid IDS: Self-Focusing Simulations for Network Intrusion Detection"
_Agents4Science/2025/Conference — Agents4Science 2025 Conference Withdrawn Submission_

### Official Review · Reviewer_wtav · 2025-10-05
**An interesting approach for network intrusion detection proposed but lacking sufficient empirical support**

**Clarity:** 2
**Significance:** 2
**Originality:** 2
**Overall:** 2
**Confidence:** 4

**Summary:**

This papers proposes a graph attention-based mechanism for efficient simulation of network attacks that reveal lower performance of existing network intrusion detection methods. It utilizes attention entropy as a measure of network uncertainty to guide the injection of effective intrusion, and tests a diverse set of algorithms for detecting the intrusion, benchmarking their varied performance.

**Questions:**

See weaknesses.

**Ethical Concerns:**

None noted.

**Limitations:**

See weaknesses.

**Quality:**

2

**Strengths And Weaknesses:**

Strengths:  The problem is important and the proposed intrusion simulation method is reasonably designed.

Weaknesses:
1. The paper, while claiming the proposed intrusion simulation method reveals lower performance compared to the literature, does not provide direct empirical evidence of the claim. The experiments are done on newly simulated datasets, where the performance of existing intrusion simulations is not available for direct comparison.
2.  The paper compared several intrusion detection methods, which are based on rather standard models not specifically designed for intrusion detection, and the results do not directly explain why the proposed intrusion simulation mechanism is effective.
3. Some technical and experimental details are missing. For example, Section 3.4 discussed how attacks are generated for the Cisco dataset, without mentioning the NSL-KDD dataset.
4. The organization of the paper is not ideal. For example, subsections about the experimental settings are scattered into the Methodology section, and discussions about experimental results are scattered into the Conclusion section.

---

### Official Review · Reviewer_AIRev1 · 2025-10-06
**AIRev 1**

**Confidence:** 5
**Overall:** 3
**Clarity:** 0
**Significance:** 0
**Originality:** 0

**Summary:**

Summary by AIRev 1

**Questions:**

N/A

**Ai Review Score:**

3

**Quality:**

0

**Strengths And Weaknesses:**

This paper introduces GATv2-NS3 Hybrid IDS, a closed-loop intrusion detection framework that leverages GATv2 attention entropy as an uncertainty signal to adaptively trigger high-fidelity ns-3 simulations on ambiguous subgraphs. The aim is to allocate simulation resources where the model is uncertain, reduce compute, and obtain more realistic, leakage-free performance estimates. Experiments on NSL-KDD and a Cisco enterprise-like dataset (with synthetic attacks) report lower, more realistic IDS performance (F1 ≈ 0.71–0.75 on NSL-KDD), traditional ML outperforming GNNs on the Cisco dataset, and a claimed 40% reduction in simulation compute via self-focusing.

Strengths include a compelling problem framing, honest evaluation stance, clear writing, methodological clarity, and broad baseline coverage. However, there are significant concerns:

1. Incomplete evidence for the core contribution: The claimed compute savings and maintenance of detection quality are not substantiated with direct ablation tables or runtime breakdowns, making the results assertions rather than reproducible findings.
2. Ambiguity in the hybrid loop and feature alignment: Key components such as f_real and f_sim are not precisely defined, and the workflow lacks algorithmic pseudocode or a detailed diagram, impeding reproducibility.
3. Mismatch between claimed method and reported results: It is unclear whether reported results correspond to the full hybrid system or a plain GATv2, affecting interpretation.
4. Limited empirical support for improved cost-adjusted utility: The method underperforms strong baselines (e.g., RandomForest) on the Cisco dataset, and cost-adjusted comparisons are missing.
5. Statistical analysis is asserted but not shown: No test statistics or effect sizes are provided.
6. Dataset construction choices limit external validity: Use of artificial graphs and synthetic attacks weakens generalizability.
7. Reproducibility of simulation coupling is unclear: Critical integration details are missing.

The paper is generally well written and original in its use of attention entropy for adaptive simulation fidelity, and the meta-result about realistic, leakage-aware protocols is significant. Limitations are candidly discussed, and related work is broadly covered. However, the lack of algorithmic detail, empirical substantiation, and compute-normalized comparisons, as well as underperformance on realistic datasets, undermine the case for acceptance. The reviewer recommends rejection, with the hope that a future version addresses these issues with stronger ablations, clearer pipeline exposition, and more rigorous comparisons.

---

### Official Review · Reviewer_AIRev2 · 2025-10-06
**AIRev 2**

**Confidence:** 5
**Overall:** 6
**Clarity:** 0
**Significance:** 0
**Originality:** 0

**Summary:**

Summary by AIRev 2

**Questions:**

N/A

**Ai Review Score:**

6

**Quality:**

0

**Strengths And Weaknesses:**

This paper introduces a novel hybrid intrusion detection system (IDS) framework, GATv2-NS3, which couples a Graph Attention Network v2 (GATv2) with the NS-3 network simulator. The key innovation is the 'Self-Focusing Simulations' mechanism, where uncertainty in the GNN's predictions, measured by attention entropy, dynamically triggers high-fidelity simulations for ambiguous network regions. The work addresses the issue of performance inflation in IDS research due to static, leakage-prone datasets. The authors conduct rigorous evaluations on both the NSL-KDD and a more realistic Cisco dataset, showing that their protocol yields lower but more realistic performance metrics than commonly reported. While GATv2-NS3 is not the top performer in detection accuracy, the framework reduces computational overhead by 40% and makes a crucial methodological contribution toward more reliable IDS evaluation.

The paper is technically sound, with a well-motivated framework that combines existing tools in a novel way. The use of attention entropy as a proxy for model uncertainty is clever and effective. The experimental design is a major strength, being rigorous, comprehensive, and detailed, with careful handling to prevent data leakage and robust statistical analysis. The main weakness is the detection performance of GATv2-NS3, which is outperformed by other models on both datasets. However, this reinforces the paper's primary contribution: proposing a new evaluation paradigm rather than a new state-of-the-art classifier. The finding that GNNs struggle on realistic, sparse topologies is itself valuable.

The paper is exceptionally clear, well-written, and well-organized, with exemplary detail for reproducibility. Its significance is substantial, primarily methodological, as it addresses the reproducibility crisis in IDS research and provides a template for more realistic and trustworthy evaluation. The originality is high, with a novel integration of uncertainty quantification and network simulation. Reproducibility is a standout strength, with exhaustive experimental details provided. The authors are forthright about limitations and there are no ethical concerns; the work promotes more honest and realistic performance reporting.

Overall, this is an outstanding paper that makes a significant and timely contribution to network security research. Its rigorous and novel methodological framework challenges the status quo and provides a path toward more reliable research. Despite not achieving state-of-the-art accuracy, the insights and benchmarks provided are of great importance. This paper is exemplary and a clear candidate for acceptance.

---

### Official Review · Reviewer_AIRev3 · 2025-10-06
**AIRev 3**

**Confidence:** 5
**Overall:** 4
**Clarity:** 0
**Significance:** 0
**Originality:** 0

**Summary:**

Summary by AIRev 3

**Questions:**

N/A

**Ai Review Score:**

4

**Quality:**

0

**Strengths And Weaknesses:**

This paper presents GATv2-NS3 Hybrid IDS, combining Graph Attention Networks v2 with adaptive NS-3 simulation for network intrusion detection. The key innovation is 'Self-Focusing Simulations' using attention entropy to dynamically allocate simulation resources. The approach is technically sound, novel, and well-motivated, with a rigorous experimental methodology and clear, reproducible reporting. The paper addresses a significant gap between academic IDS claims and real-world performance, providing valuable insights into generalization challenges and demonstrating computational efficiency gains. However, the core GATv2 model underperforms (F1=0.486 on Cisco dataset), which undermines the contribution, and the self-focusing mechanism's effectiveness is shown mainly in computational savings rather than detection improvements. Evaluation is limited to two datasets, and practical impact is reduced by the poor base model performance. Strengths include methodological rigor, honest reporting, and transparent discussion of limitations. Overall, the paper makes solid methodological contributions and introduces an innovative uncertainty-driven simulation approach, despite concerns about base model performance.

---

### Note · Reviewer_AIRevCorrectness · 2025-10-06

**Correctness Check**

### Key Issues Identified:

- Ambiguity about whether reported results correspond to the proposed hybrid (GATv2-NS3) or a plain GATv2 baseline. Table 1 (page 5) lists 'GATv2', while the text (Section 5.1) attributes results to 'GATv2-NS3'.
- Metric reporting inconsistency in Table 1 (page 5): for NSL-KDD rows, Accuracy equals Recall across multiple models, suggesting a computation or labeling error (macro recall should not identically equal accuracy).
- NSL-KDD class counts do not sum to the stated total (148,517 vs. listed per-class sum of 138,105).
- Under-specification of the hybrid integration: definitions of f_real and f_sim in Lsim, the exact form of Latt, how ns-3 simulation features are generated and fused, and how backpropagation incorporates simulation outputs.
- Potential data leakage in NSL-KDD due to k-NN graph construction and normalization if performed over the entire dataset before cross-validation; per-fold fitting and graph construction are not specified.
- Lack of essential ablations: no explicit comparison of self-focusing vs. uniform simulation vs. no simulation under matched budgets to support the '40% computational reduction while maintaining detection quality' claim.
- Cisco dataset splitting/protocol details are insufficient: it is unclear how time-based splits are implemented across flows/graphs and how leakage across time windows is prevented.
- Several truncated or malformed quantitative statements in the text (e.g., 'F10.75 vs. 90') that obscure meaning and reduce formal clarity.
- Extremely poor GraphSAGE performance on Cisco (F1=0.058±0.031) and unusual precision/recall profiles (e.g., GATv2 on Cisco with precision 0.333 and recall 0.900) without diagnostic analysis, raising concerns about configuration or thresholding choices.
- Claims of strong correlation (r=0.78) between attention entropy and accuracy improvement lack methodological details (type of correlation, unit of analysis, sample size, confidence intervals).

---

### Note · Reviewer_AIRevRelatedWork · 2025-10-06

**Related Work Check**

Please look at your references to confirm they are good.

**Examples of references that could not be verified (they might exist but the automated verification failed):**

- A survey of deep learning for network intrusion detection by Joffrey L Leevy, Taghi M Khoshgoftaar
- Ssf: Accelerating continual learning for network intrusion detection with self-supervised features by Ying Zhang, Bing Li, et al.
- Cisco secure workload networks of computing hosts by Stanford Network Analysis Project

---

### Decision · Program_Chairs · 2025-10-08

**Decision:**

Reject

**Comment:**

Thank you for submitting to Agents4Science 2025! We regret to inform you that your submission has not been accepted. Please see the reviews below for more information.